# Postpartum blood pressure self-management following hypertensive pregnancy: protocol of the Physician Optimised Post-partum Hypertension Treatment (POP-HT) trial

Jamie Kitt [1], Annabelle Frost,[1] Jill Mollison,[2] Katherine Louise Tucker [2], Katie Suriano,[1] Yvonne Kenworthy,[3] Annabelle McCourt,[1] William Woodward,[1] Cheryl Tan,[1] Winok Lapidaire,[1] Rebecca Mills,[4] Miriam Lacharie,[4] Elizabeth M Tunnicliffe,[4] Betty Raman,[4] Mauro Santos,[5] Cristian Roman,[5] Henner Hanssen,[6] Lucy Mackillop,[7] Alexandra Cairns,[7] Basky Thilaganathan [8], Lucy Chappell,[9] Christina Aye,[7] Adam J Lewandowski,[10] Richard J McManus [11], Paul Leeson[1]

For numbered affiliations see end of article.

**Correspondence to**
Dr Jamie Kitt;
jamie.kitt@cardiov.ox.ac.uk

## ABSTRACT

**Introduction** New-onset hypertension affects approximately 10% of pregnancies and is associated with a significant increase in risk of cardiovascular disease in later life, with blood pressure measured 6 weeks postpartum predictive of blood pressure 5–10 years later. A pilot trial has demonstrated that improved blood pressure control, achevied via self-management during the puerperium, was associated with lower blood pressure 3-4 years postpartum. Physician Optimised Post-partum Hypertension Treatment (POP-HT) will formally evaluate whether improved blood pressure control in the puerperium results in lower blood pressure at 6 months post partum, and improvements in cardiovascular and cerebrovascular phenotypes.

**Methods and analysis** POP-HT is an open-label, parallel arm, randomised controlled trial involving 200 women aged 18 years or over, with a diagnosis of pre-eclampsia or gestational hypertension, and requiring antihypertensive medication at discharge. Women are recruited by open recruitment and direct invitation around time of delivery and randomised 1:1 to, either an intervention comprising physician-optimised self-management of postpartum blood pressure or, usual care. Women in the intervention group upload blood pressure readings to a 'smartphone' app that provides algorithm-driven individualised medication-titration. Medication changes are approved by physicians, who review blood pressure readings remotely. Women in the control arm follow assessment and medication adjustment by their usual healthcare team. The primary outcome is 24-hour average ambulatory diastolic blood pressure at 6–9 months post partum. Secondary outcomes include: additional blood pressure parameters at baseline, week 1 and week 6; multimodal cardiovascular assessments (CMR and echocardiography); parameters derived from multiorgan MRI including brain and kidneys; peripheral macrovascular and microvascular measures; angiogenic profile measures taken from blood samples and levels of endothelial circulating and cellular biomarkers;

## Strengths and limitations of this study

► Physician Optimised Post-partum Hypertension Treatment (POP-HT) is the first randomised trial that is powered to detect whether postpartum blood pressure self-management can significantly improve blood pressure control at 6–9 months post partum, in women affected by new onset hypertension in pregnancy.

► POP-HT will also be the first randomised trial to investigate whether this improved blood pressure control translates into beneficial cardiovascular, vascular and cerebrovascular modelling, and will help elucidate the mechanisms behind adverse remodelling.

► The technology used to facilitate self-management in the study is readily translatable into widespread clinical practice, once it has been subject to the relevant regulatory approval and further validation .

► The risk of drop-out, amplified by the COVID-19 global pandemic, could affect the ability to remain adequately powered to test the secondary outcome measures of the trial.

► The trial requires high levels of participant motivation and engagement during a busy time of the participants' lives. Patients in the intervention arm are required to submit daily readings when on treatment and adjust their medication as instructed by the app. Weekly readings are required thereafter for the duration of the study. This is in addition to the two periods of 24 hours ABPM over the 6-month period. The degree of engagement required may limit recruitment into the trial and/or result in high drop-out/withdrawal rate.

and objective physical activity monitoring and exercise assessment. An additional 20 women will be recruited after a normotensive pregnancy as a comparator group for endothelial cellular biomarkers.

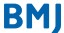

**Ethics and dissemination** IRAS PROJECT ID 273353. This trial has received a favourable opinion from the London—Surrey Research Ethics Committee and HRA (REC Reference 19/LO/1901). The investigator will ensure that this trial is conducted in accordance with the principles of the Declaration of Helsinki and follow good clinical practice guidelines. The investigators will be involved in reviewing drafts of the manuscripts, abstracts, press releases and any other publications arising from the study. Authors will acknowledge that the study was funded by the British Heart Foundation Clinical Research Training Fellowship (BHF Grant number FS/19/7/34148). Authorship will be determined in accordance with the ICMJE guidelines and other contributors will be acknowledged.
**Trial registration number** NCT04273854.

## INTRODUCTION

Hypertensive disorders of pregnancy affect 10% of pregnancies, which equates to >80 000 women per year in the UK.[1] One study showed that 50% of women with pre-eclampsia have persistent significant hypertension on day five following delivery,[2] with blood pressure (BP) control remaining an issue for up to 6 weeks after childbirth, often requiring multiple medications and careful titration. At the same time, competing demands on mothers, not least from her newborn baby, are associated with poor adherence and/or poor levels of clinical contact. Drug titration can, therefore, be sporadic exacerbating poorly controlled hypertension. The risk of poor BP control during this period may extend into later life. Hypertensive disorders of pregnancy are associated with a twofold increase in risk of subsequent cardiovascular disease[3 4] with a third presenting with chronic hypertension within 10 years.[5] We observed that high BP during the first 6 weeks post partum is related to the risk of higher BP in the 5–10 years following a hypertensive pregnancy.[6] Furthermore, the Self-management of Postnatal Anti-hypertensive Treatment (SNAP-HT) randomised controlled pilot study[7] showed BP control in the post-partum period can be improved through self-management, with a mean improvement of 5.2 mm Hg in systolic and 6 mm Hg in diastolic BP at 6 weeks post partum. However, the most striking finding was that mean diastolic BP remained 4.5 mm Hg lower in the intervention group at 6 months' post partum; even after all but two mothers had stopped medication. Long-term follow-up of the women involved in the trial demonstrated diastolic BP remained significantly lower in the intervention group (−6.8 mm Hg), at more than 3 years post partum, even when adjusted for 'lifestyle risk factors'.[8] In those below the age of 50 years diastolic BP is also a better correlate to long-term cardiovascular risk and is the predominant pathophysiology in young adults[9] before progression to the mixed and systolic patterns of hypertension seen in those over 50.[9] Given all our patients are young females, below the age of 50 years, 24 hours overall average diastolic BP was selected as the primary outcome for our trial.

In the general population, high BP is strongly related to long-term risk of cardiovascular disease and is the leading risk factor for loss of disability-adjusted life-years in high-income and low-middle-income countries. Every 10 systolic/5 diastolic mm Hg of BP reduction associates with a ~40% reduction in lifetime stroke risk and ~20% reduction in coronary heart disease risk.[10] If the difference observed in SNAP-HT could be translated to all women who have a hypertensive pregnancy, significant reductions in disease burden could be achieved in the population. However, several questions remain to determine the potential clinical translational benefits of self-management of BP in the postpartum period. First, can a similar reduction in BP be achieved with updated technology in a trial powered to detect diastolic BP differences at 6 months? Bluetooth-enabled home BP monitors allow for automated upload of readings to the app, reducing the need for manual entry during a busy period in the patients' lives. This also facilitates telemonitoring by physicians who can review and advise on readings.[11–13] Second, does improved postpartum BP control also result in reduced end organ damage in the cardiac, vascular and cerebrovascular systems? Hypertensive pregnancies are associated with early changes in cardiac, vascular and brain structure and function, which are disproportionate to their cardiovascular risk profile.[14–16] This may explain why this population have an increased risk of later cardiovascular and cerebrovascular disease. It is possible these changes may emerge during pregnancy and persist long term, independent of postpartum BP variability, although the significant cardiovascular adaptations that emerge during pregnancies complicated by hypertensive disorders of pregnancy are known to reverse to some extent during the postpartum period.[17–20] An alternative hypothesis it that the long-term risk reflects a failure of the cardiac, vascular or cerebral systems to 'normalise' after pregnancy.[21 22] If so, improved postpartum BP control may offer a new approach to modify these long-term end-organ changes, in addition to any beneficial effects on BP.

## HYPOTHESES

The primary hypothesis is that BP self-management with clinician oversight will reduce diastolic BP at 6–9 months post partum; in women requiring antihypertensive medication in the puerperium after a hypertensive pregnancy.

The secondary hypotheses are that BP lowering in the puerperium and strict control within predefined target ranges during this time period will improve cardiovascular, and cerebrovascular and vascular phenotypes in this cohort including:

► MRI indices of cardiovascular structure and function.
► MRI indices of cerebral perfusion, white matter integrity, subcortical volumes.
► MRI assessment of aortic compliance.
► Echo measures of cardiovascular structure and function, especially diastolic function and left atrial volume.
► Improved cardiovascular adaptation to exercise, assessed by exercise ejection fraction during cardiopulmonary exercise testing (CPET).

In addition, it is hypothesised the intervention will lead to improvements in peripheral vascular function, including integrity of the retinal vasculature and markers of vascular stiffness. The tertiary hypotheses are that self-management will be associated with higher quality of life scores once discharged from hospital, fewer postnatal readmissions to hospital, improved endothelial function; and fewer signs of kidney injury and fibroinflammation.

## METHODS

Planned trial period: 31 December 2019–01 December 2030.

Planned Recruitment period: 31 December 2019–31 August 2021.

### Study design

Physician Optimised Post-partum Hypertension Treatment (POP-HT) is a single-centre, open-label, two-arm parallel, randomised controlled trial in women who develop hypertensive disorders of pregnancy, who require antihypertensive treatment at the time of discharge. This study will investigate the effectiveness of postpartum physician assisted self-management of BP compared with standard care over the first 6 months post partum.

We will recruit 200 participants, who will be randomly allocated in a 1:1 fashion to either the intervention or control arm. The intervention arm will comprise app-based home BP monitoring (including periods of home 24 hours ambulatory blood pressure monitoring (ABPM) coupled with physician-assisted self-management. The control arm will receive 'standard' levels of National Health Service (NHS) care from their GP and midwives and health visitors. All participants will be recruited from the Oxford Women's Centre at the John Radcliffe Hospital, which sees approximately 25 patients with hypertensive disorders of pregnancy per month. A trial flow chart is presented in online supplemental appendix A and a schedule of procedures in online supplemental appendix B. The expected duration of participant involvement will be up to a maximum of 12 months.

### Endothelial cell substudy

The purpose of the sub study of 20 women is to provide a reference population of women not affected by hypertensive disease. In the sub study, 20 healthy postnatal women will undergo measurements of specific characteristics of blood cells and circulating factors involved in inflammation and endothelial dysfunction. This small subcohort population will validate how blood cells and circulating factors vary naturally and may be affected by external factors such as mode of delivery and BP. These 20 normotensive participants will be compared with 20/200 participants who have had a hypertensive pregnancy and that are in the main study. These 20/200 will provide additional consent for blood sampling for endothelial cells at baseline and at the V4 visit. The 20 normotensive patients will be recruited from the postnatal ward in the Oxford Women's Centre and the normotensive participants will be recruited directly to the substudy and, not be expected to participate in the main POP-HT study (see online supplemental appendix C for a schedule of procedures for the substudy).

### Study intervention

The intervention is the provision of a wireless Bluetooth enabled OMRON® Evolv BP monitor, validated for widespread clinical use, including in pregnancy,[23] alongside the installation of a proprietary smartphone app. The app will assist the self-management of postpartum BP, with physician oversight of medication adjustment. Following the baseline visit, those randomised to the intervention arm will be provided with an OMRON® Evolv monitor and the app installed on the participant's phone and its use demonstrated. The participants will have ample time to become familiar with the device and app prior to discharge, as in SNAP-HT.[4] They will also be provided with an 'intervention-arm information sheet,' which contains information about the self-monitoring process as well as a frequently asked questions section and contact number for any technical issues. The system was developed by the Oxford Institute of Biomedical engineering who have successfully developed apps for several other BP studies including SNAP-HT,[7] TASMIN[24] and BUMP (NCT03334149). Participants in the intervention arm will be asked to start home BP readings on the day of discharge. Figure 1 illustrates the method of BP self-management.

Following discharge from hospital, participants will be asked to upload their home BP readings in a standardised manner. These readings are in turn uploaded to the secure NHS hosted web-based platform. The readings are automatically cross-checked against predefined algorithms and an appropriate notification will be generated in response.

The medications prescribed to each participant at discharge will be decided by their clinical care team. The medication schedule will be uploaded to a secure NHS hosted web-based platform which syncs automatically with the app. This approach was trialled successfully in the SNAP-HT pilot trial[7] and this study will be using the same approach to develop these dose titration schedules based on discharge medication. Medication titration will be done in line with the updated National Insitute of Clinical Excellence (NICE) guidance NG133 in the intervention arm (and for those in the control arm clinicians are anticipated to also follow this new NICE guideline).[11] Down-titration is triggered when BP is consistently <130 mm Hg and <80 mm Hg diastolic and the process for this is explained in further detail in online supplemental appendix D.

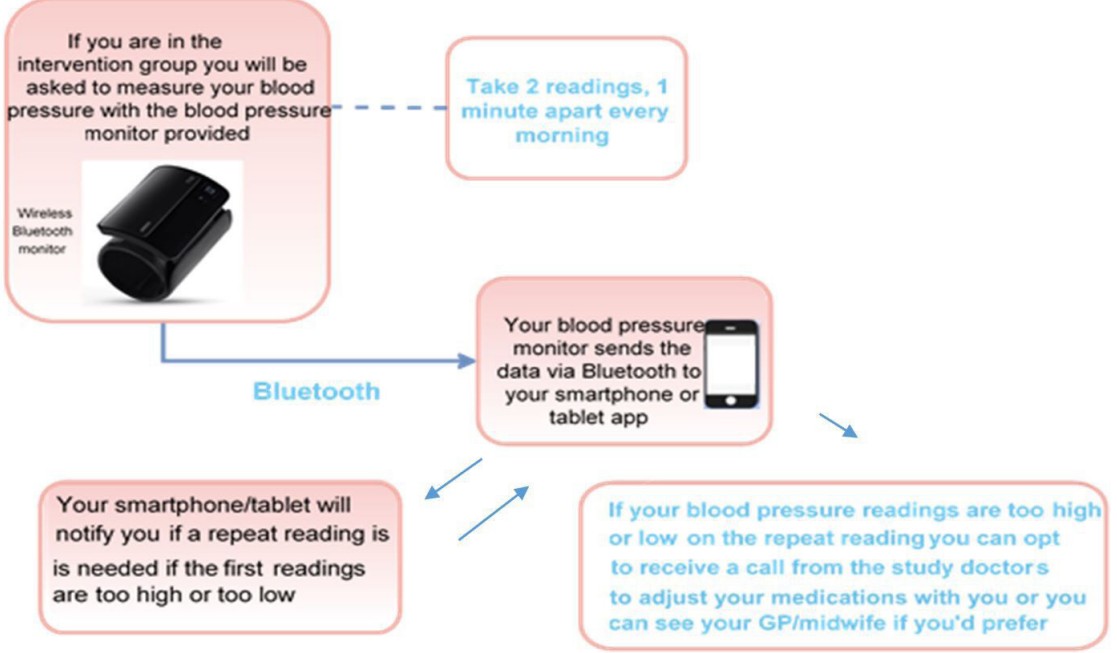

**Figure 1** Illustration of self-management of blood pressure for women randomised to the intervention. GP, general practitionerLearn to pronounce.

## Patient and public involvement

The study team hosted regular patient and public involvement (PPI) meetings prior to the study commencing to understand the experiences of patients participating in prior self-management and pre-eclampsia studies, and to assist with study design and methodology. This included consulting participants and investigators from other related studies conducted by this group. Participant-facing information documents, were reviewed by PPI members with experience of raised BP in pregnancy. PPI members also helped design the intervention to ensure it was acceptable for participants. The PPI group will also assist in the drafting of study results for dissemination to participants.

## Study aims and objectives

For clarity; the time point of the primary and BP based outcomes is 6–9 months post partum, whereas an extra 3 months (6–12 months) was approved by the trial steering committee (TSC), REC and HRA for completion of the other secondary outcome data collection (as part of an amendment to mitigate against the impact of the COVID-19 pandemic on follow-up rates for these additional measures). This explains the different time points shown in table 1.

## Eligibility and recruitment

The procedures for each study visit and the estimated time each will take are listed in online supplemental appendices B and C, respectively.

All trial participants will be recruited locally from the Women's Centre at the John Radcliffe Hospital, Oxford. Screening will be carried out by the patient's clinical care team. Consent will be performed by the research team, once verbal consent is given to the clinical team for them to be approached.

## Main trial participants

All participants will be females of childbearing aged 18 years or over . Entry into the trial will require a clinician confirmed diagnosis of either gestational hypertension or pre-eclampsia defined by NICE NG 133,[11] that requires antihypertensive medication.

## Inclusion criteria

► Participant is willing and able to give informed consent for participation in the trial.
► Female, aged 18 years or above.
► Clinician confirmed diagnosis of either gestational hypertension or pre-eclampsia defined by NICE NG 133 and remains in hospital after delivery.
► Requiring antihypertensive medication at the point of discharge from secondary care.
► Participant has clinically acceptable laboratory results and clinical course post partum with no other adverse complicating factor requiring prolonged admission post partum that would make participation unfeasible as judged by the CI. Examples would include stroke sequalae, ongoing DIC or other significant life-threatening comorbidity
► In the investigator's opinion, is able and willing to comply with all trial requirements including ownership of a smartphone or tablet and willing to use the smart-phone app if randomised to that arm.
► Sufficient competence in English language to follow the app instructions and partake in the study, as judged by the CI.

**Table 1** List of study aims and objectives with respective timepoint (s) for the measurement of each objective

| | Objectives | Outcome measures | Timepoint(s) |
|---|---|---|---|
| Primary | To compare postpartum diastolic BP in the intervention arm vs the control arm. | 24-hour average diastolic BP measured by SPACELAB 90217 24 hours ABPM | Visit 4 (6–9 months post partum) |
| Secondary | To compare the effect of the intervention on cardiovascular, cerebrovascular and vascular phenotypes | **BP based**<br>▶ 24-hour average systolic blood pressure assessed by SPACELAB 90217 24-hour ABPM<br>▶ Mean diurnal diastolic blood pressure assessed by SPACELAB 90217 ABPM<br>▶ Mean diurnal systolic blood pressure assessed by SPACELAB 90217 ABPM<br>▶ Mean nocturnal diastolic blood pressure assessed by SPACELAB 90217 24-hour ABPM<br>▶ Mean nocturnal systolic blood pressure assessed by SPACELAB 90217 24-hour ABPM<br>▶ Mean bedside diastolic blood pressure measured during study visit (mean of 2+3)<br>▶ Mean bedside systolic blood pressure measured during study visit (mean of 2+3) | 6-9 months for the 24-hour ABPM measures<br><br>Baseline, week 1, week 6 and 6–9 months for the bedside BP measures |
| | | **Cardiac MRI**<br>▶ Left ventricular (LV) mass indexed to end-diastolic volume and body surface area (BSA)<br>▶ LV EDV indexed to BSA<br>▶ LV wall thickness<br>▶ LA volume indexed to BSA<br>▶ Right ventricular (RV) mass indexed to end-diastolic volume and body surface area<br>▶ RV EDV indexed to BSA<br>▶ RA volume indexed to BSA<br>▶ LV ejection fraction (EF) and RVEF<br>▶ LV and RV stroke volumes indexed to BSA<br>▶ Myocardial fibrosis<br>▶ ECV (extracellular volume) | For cardiac MRI at 6–12 months post partum |
| | | **Echo**<br>▶ LV Diastolic function:E/E' average, E/A ratio, E deceleration time<br>▶ Global longitudinal strain (GLS)<br>▶ LV systolic function (EF by Biplane Simpson's)<br>▶ LA volume by Biplanar assessment | At baseline and at 6–12 months post partum for echo outcome measures |
| | | **Vascular**<br>▶ Pulse wave velocity<br>▶ Augmentation index and Aortic BP<br>▶ Aortic distensibility (MRI) | PWV, Aortic BP and AI at baseline and at 6–12 months (aortic stiffness); aortic compliance (on MRI) at 6–12 months |
| | | **Cerebrovascular**<br>▶ Total white matter hyperintensity volume<br>▶ Cerebral blood flow<br>▶ Mean vessel thickness of the middle and posterior cerebral arteries and internal carotid artery | 6–12 months post partum for all Brain MRI measures. |
| | | **Retinal**<br>▶ The corrected central retinal arteriolar equivalent<br>▶ The corrected central retinal venular equivalent<br>▶ Corrected central retinal arteriolar equivalent/corrected central retinal venular equivalent ratio. | 6–12 months post partum for all retinal measures |
| Tertiary | | **Exercise echo**<br>Exercise ejection fraction (echo) and exercise LA volume at 50% of peak workload during a bicycle cardiopulmonary exercise test (CPET) | 6–12 months post partum |
| | | **CPET**<br>VO2 at VT1 | 6–12 month post partum |
| | To explore in vitro vascular function in a substudy of 20 women | Assessment of endothelial cell function and circulating biomarker levels associated with vascular angiogenesis and inflammation in normotensive and hypertensive women to determine if BP improvement can affect vascular function | From baseline to 6–12 months post partum |
| | | T1 mapping of the kidneys to look at cortico-medullary differentiation | 6–12 months post partum |

**Table 1** Continued

|  | Objectives | Outcome measures | Timepoint(s) |
|---|---|---|---|
|  | To explore presence/absence of kidney injury and fibro-inflammatory status | EQ-5D-5L health questionnaire results | Baseline, week 1, week 6 and 6–12 months post partum |
|  | Quality of life assessment | Qualitative semistructured interviews in a subset of individuals as well as assessment of acceptability and feasibility within the intervention arm | 6–12 months post partum |
|  | Participant experience: assessment of individual experience following intervention | Readmission number in each arm | 0–12 months post partum |
|  | No of readmissions in intervention versus control arm | Number and frequency of side effects reported (intervention via the app and control during follow up calls | 0–12 months post partum |
|  | Side effect impact |  |  |
| Intervention(s) | The intervention will consist of physician-optimised self-management of post-partum BP. Women will follow a 'smartphone' app-based algorithm for medication titration, which will provide individualised dose titration advice. This is overseen and any change is approved by physicians who review the uploaded readings and respond to telemonitored abnormal readings in a timely fashion. |  |  |
| Comparator | The control arm will be managed as per usual NHS-led care with assessment by their own healthcare professionals and adjustment of medications as required. The BP of this group will be monitored and recorded at the same time-points and in the same manner as the intervention arm as with all other secondary outcome measures. |  |  |

ABPM, Ambulatory Blood Pressure Monitoring ; AI, Augmentation Index; BP, blood pressure; EDV, End Diastolic Volume; EQ-5D-5L, The descriptive system comprises five dimensions: mobility, self-care, usual activities, pain/discomfort and anxiety/depression to assess quality of life; NHS, National Health Service; RA, Right atrial; RWT, Relative Wall thickness.

## Exclusion criteria (the participant may not enter the trial if any of the following apply)

► Significant renal or hepatic impairment that would affect safe medication titration and adjustment as part of the trial, as deemed by the investigator.
► Participant with life expectancy of less than 6 months.
► Any other significant disease or disorder which, in the opinion of the investigator, may either put the participant at risk through participation in the trial, influence the result of the trial, or impair the participant's ability to participate in the trial.
► Participants who have participated in another research trial involving an investigational product in the past 12 weeks.
► Women with pre-existing hypertension will be excluded, as this is a separate pathology that would affect the efficacy of the study intervention and affect the primary and secondary outcomes of the study.

## ENDOTHELIAL CELLS SUBSTUDY

Dysregulation of the vascular endothelium and endothelial cells has been observed in the pathogenesis and progression of several cardiovascular diseases, including hypertension and hypertensive pregnancy disorders.[25–31] Studies have demonstrated that significant peripartum inflammation, endothelial dysfunction and angiogenic imbalance extends beyond delivery, and persists to 5–10 years post partum.[32–34] Endothelial colony-forming cells (ECFCs) represent a highly proliferative subtype of endothelial progenitor cells, which play a vital role in the regulation of vascular homeostasis.[35–37] Herein, we will investigate endothelial cell function using peripheral blood derived-ECFCs and

angiogenic biomarkers in the blood at baseline and at the final visit (V4), which takes places approximately 6 months post partum.

The inclusion criteria for the blood validation substudy include:

► Participant is willing and able to give informed consent for participation in the trial.
► Female, aged 18 years or above.
► Normotensive (BP <140/90 mm Hg) throughout antenatal and postnatal period (except the 20/200 recruited from the main study).

The exclusion Criteria for POP-HT blood validation substudy include:

► A hypertensive disorder of pregnancy (for the 20 normotensive recruits required).
► Use of beta blockers such as atenolol or equivalent.
► Body mass index (BMI) >35 kg/m$^2$.
► Evidence of cardiomyopathy, inherited cardiac conduction abnormalities, congenital heart disease or significant chronic disease relevant to cardiovascular status.
► Folic acid or folate supplementation in the third trimester.

## Randomisation

Randomisation will be performed as soon as possible following the baseline visit. Randomisation will be carried out using a secure web-based randomisation software (embedded within Castor®). Participants will be randomised on a 1:1 basis and two minimisation factors will be used to ensure that the groups are matched as well as possible:

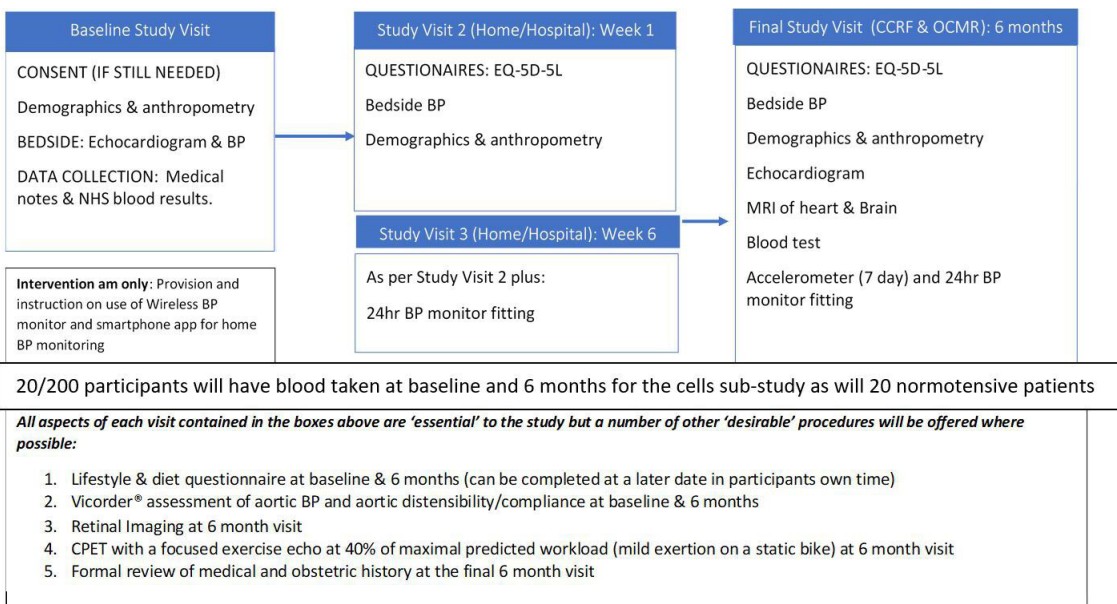

| Baseline Study Visit | Study Visit 2 (Home/Hospital): Week 1 | Final Study Visit (CCRF & OCMR): 6 months |
|---|---|---|
| CONSENT (IF STILL NEEDED) | QUESTIONAIRES: EQ-5D-5L | QUESTIONAIRES: EQ-5D-5L |
| Demographics & anthropometry | Bedside BP | Bedside BP |
| BEDSIDE: Echocardiogram & BP | Demographics & anthropometry | Demographics & anthropometry |
| DATA COLLECTION: Medical notes & NHS blood results. | | Echocardiogram |
| | | MRI of heart & Brain |
| | **Study Visit 3 (Home/Hospital): Week 6** | Blood test |
| **Intervention am only**: Provision and instruction on use of Wireless BP monitor and smartphone app for home BP monitoring | As per Study Visit 2 plus: 24hr BP monitor fitting | Accelerometer (7 day) and 24hr BP monitor fitting |

20/200 participants will have blood taken at baseline and 6 months for the cells sub-study as will 20 normotensive patients

*All aspects of each visit contained in the boxes above are 'essential' to the study but a number of other 'desirable' procedures will be offered where possible:*

1. Lifestyle & diet questionnaire at baseline & 6 months (can be completed at a later date in participants own time)
2. Vicorder® assessment of aortic BP and aortic distensibility/compliance at baseline & 6 months
3. Retinal Imaging at 6 month visit
4. CPET with a focused exercise echo at 40% of maximal predicted workload (mild exertion on a static bike) at 6 month visit
5. Formal review of medical and obstetric history at the final 6 month visit

**Figure 2** Flow chart of proposed study visits. BP, blood pressure. Cardiovascular Clinical Research Facility (CCRF), Oxford Centre for Magnetic Resonance (OCMR)

► Primary factor: Gestational age at the time of presentation with pre-eclampsia/gestational hypertension (agreed on as a surrogate marker of disease severity).

► Secondary factor: Prescription of ACE inhibitor (Enalapril) at the time of randomisation.

The full trial protocol (available on request) details more information on allocation concealment and implementation of the randomisation.

### Assessments during study visits

A flow chart of the proposed study visits is included below (figure 2). All data will be recorded directly into CASTOR® electronic data capture forms where possible in real time. Any data requiring postprocessing will be entered into CASTOR® following such analysis. During the study visits, all procedures are performed on participants in both the intervention and control arms with the exception of provision of the intervention; which is reserved only for those randomised to that arm. At all study visits involving a review of the medical and obstetric history, the number and dose(s) of antihypertensive medication is recorded to allow comparison between groups post hoc.

As figure 2 illustrates, there will be 'essential' components to each study visit. However, as the research is being carried out on women with newborn babies, some components have been classed as 'desirable' to allow for shortening of the study visits, if required, without affecting the primary outcome. A number of modifications to the original study design, and the means of performing the above assessments, have also been made to mitigate the impact of the COVID-19 global pandemic (please see online supplemental appendix E for further detail . All amendments have been submitted to; and approved by SPONSOR, the REC and HRA, the local

hospital (OUH) trial management authority (TMA); and other relevant parties have been notified where needed (see online supplemental appendix F for amendment history).

### Baseline visit (week 0)
#### Demographics and anthropometry
Assessment will include recording of the antenatal booking height, weight and BMI (obtained from notes) and the mid-left arm circumference.

#### Bed-side BP measurement
Participants will have their BP checked after 5 min' rest using the automated mode of a validated sphygmomanometer. Three BP readings will be taken at intervals of 1 min. The measurement technique advised by the British Heart Foundation and NICE NG133[11] will be strictly followed.

#### Echocardiogram (cardiac ultrasound) scan
Cardiac ultrasound imaging will be performed by a trained sonographer to evaluate cardiac structure and function. British Society of Echocardiography guidelines will be followed for collection of a standard clinical imaging dataset.

#### Collate data from medical notes and review blood results
A study team member will review the medical notes (paper and electronic) to document relevant medical history as stated in the full trial protocol.

#### Quality of Life questionnaire (EQ-5D-5L)
Participants will be provided with an EQ-5D-5L questionnaire via email.

*Desirable: Vicorder (vascular measures and central BP)*

Resting measures of vascular stiffness including pulse wave velocity and central BP will be collected using a non-invasive device (Vicorder).

*Desirable: lifestyle and diet questionnaire*

The questionnaire, sent via email, combines validated questions piloted or used in previous studies. Information will be collected on factors that affect BP including: smoking frequency, alcohol and salt intake, exercise and family history.

**Intervention provision: automated BP cuff provided and POP-HT app installed (those randomised to intervention arm only)**

At the end of the baseline visit, those individuals that are randomised to the intervention arm will be issued with an OMRON EVOLV automated BP cuff (validated for use in pregnancy[23]). They will also download the POP-HT smartphone app and their use will be demonstrated. The participant will then have the remainder of their stay in hospital to practice. This is to ensure all parties are confident and competent prior to discharge home, at which point the intervention will start. Participants will be able to contact the study team via telephone or email for any technical problems.

Control arm during COVID-19

During the COVID-19 pandemic, the Royal College of Obstetricians and Gynaecology (RCOG) recommends 'self-monitoring 2–3 times in the first week after discharge' for women who have had a hypertensive pregnancy. Therefore, those women allocated to the control arm, who are unable to obtain a monitor from the Oxford Women's centre/NHS service, will be provided with a validated BP home-monitoring device by the trial team to ensure they can adhere to this RCOG guidance during week one. These monitors will be provided to enable the control arm participants to monitor their own BP and in turn liaise with their own GP/mid-wife to adjust their management based on their readings. The study team will not be offering remote management to the control arm or providing them with an app, interpretation of the readings and management decisions are to be taken by their own GPs/clinicians. These monitors will also be used to allow remote BP measurements during the study visits at weeks 1 and 6.

Subsequent visits

► During the week 1 and 6 follow-up, the research team will perform all procedures for participants in both arms. As a result of the COVID-19 pandemic, all aspects of these follow-ups will be conducted remotely via video (and/or phone call).

Visits 2 and 3
*Weeks 1 and 6 (± 5 days) postdischarge*

BP will be measured as per the baseline visits, up-to-date anthropometry will be measured, and an ED-5D-5L questionnaire will be completed.

*Visit 3 will also involve an ambulatory BP monitor (to be worn for a 24-hour period)*

A 24-hour ambulatory BP monitoring will be initiated at the end of the study visit using a validated, calibrated, automated oscillometric, ambulatory devices (SPACELABS 90 217 or equivalent). Correct cuff size will be chosen based on arm circumference recorded at weeks 1 and 6.

Visit 4 (6 months (up to 12 months) post partum)

Participants will be invited to this final study visit. This is a more comprehensive visit with both essential and desirable components. A female chaperone will be offered and, where possible, all echocardiography will be performed by a female sonographer. In extenuating circumstances, such as COVID-19 national lockdowns, the 24 hours BP monitoring and the procedures below can be conducted remotely via video call. Otherwise, these will be performed in person as described for the baseline visit.

► Demographics and anthropometry: Assessment will be performed as outlined above.
► **BP:** Assessment will be performed as outlined above.
► Quality of Life questionnaire (EQ-5D-5L).
► Fitting of a 24-hour BP monitor: Assessment will be performed as outlined above.
► Fitting of an activity monitor: For remote video calls, the accelerometer will be preprogrammed based on participant reported height, weight and hand dominance and then posted to the participant. For visits carried out in person, the accelerometer will be programmed during the visit, based on their height and weight recorded as part of the study visit and, fitted to their non-dominant wrist.
► Review of medical and obstetric history and any medication side effects: as described above.

In cases where the above measures are performed remotely the additional procedures below, which cannot be done at remotely will be scheduled as soon as possible after, and within the 12-month time window defined in the protocol.

Echocardiogram cardiac ultrasound scan: will be performed as outlined above for baseline visit.

Vicorder (Vascular Measures and Central Blood Pressure) assessment: will be performed as outlined above, but while in the MRI scanner, in order that the central/aortic pressure can be correlated with the aortic distensibility images obtained during the MRI scan.

*Retinal imaging*

Retinal photography of the right eye (three single shot images centred on the optic disc) will be completed using a digital camera and imaging software following an established protocol.

*MRI*

A 3Tesla (3T) Siemens PRISMA scanner will be used to quantify brain structure and volume, followed by cardiac

structure and function, cardiac mapping, measurement of aortic distensibility; and T1 maps of the kidneys.

### Gadolinium contrast (optional)

Gadolinium will be offered as an additional optional component to the MRI to those women who are not breast feeding, as part of exploratory work that may feed into a larger future trial. The additional PIS explains the Gadolinium procedure and risks/benefits in more details. Separate informed consent will be obtained for those women who wish to participate in this aspect of the study prior to the MRI being performed.

### Blood

A venous blood sample (approximately 25 mL) will be taken at rest and include samples for (1) whole blood, plasma and serum lipid and inflammatory marker analysis and (2) analysis of biochemistry and metabolism.

### Cardiopulmonary exercise testing with exercise echo: desirable

Cardiac function and oxygen requirements in response to an incremental increase in workload will be measured via a CPET. The exercise protocol is a validated incremental protocol with established use in clinical and research practice. The exercise protocol is currently utilised in ongoing ethically approved studies conducted by the Division of Cardiovascular Medicine, and is performed on a stationary bike. The test commences with resting measures of spirometry. Participants will then exercise with an incrementally increasing workload up to 40%–60% of their estimated peak exercise capacity. A brief focused echo will be performed at rest and at 40% of their maximal predicted exercise intensity (while on the bike) to enable measurement of exercise ejection fraction.

### Substudy visits: circulating biomarker validation and evaluation

Twenty of the hypertensive pregnancy patients from the main trial will also have a blood test taken at baseline as part of their main study visit. For the normotensive participants, the following study procedures and visits listed below remain separate to the main study. Study procedures will be the same during both visits.

The baseline visit will be carried out on the postnatal ward, in the Women's Centre, prior to discharge. Normotensive participants will be invited back for a second visit at the John Radcliffe Hospital when they reach 6 months postpartum, and for those 20 hypertensive participants taking part in the main trial, the repeat blood test for the substudy will be performed during the main V4/final visit when they have other study blood tests performed.

### Procedures for all substudy patients include

Demographics and anthropometry, bed-side BP measurement and a blood test for analysis of biomarkers associated with inflammation, angiogenesis and endothelial activation as well as ECFCs. Blood tests will be taken, where possible, at the same time as clinically indicated venepuncture.

Further details and explanation of the assessments is contained in online supplemental appendix G: 'Copy of the PIS'

### Data analysis

Power calculations to determine adequate sample sizes for this trial are summarised in box 1 below.

### Statistical analysis plan

The statistical aspects of the study are summarised here with details fully described in a statistical analysis plan (SAP) that will be produced in due course.

### Description of statistical methods

The analysis will be carried out on the basis of intention-to-treat (ITT). This is, after randomisation, participants will be analysed according to their allocated intervention group irrespective of what treatment they actually receive. Patient demographic characteristics and other baseline information will be summarised by treatment group. Numbers (with percentages) for binary and categorical variables and mean (SD), or median (IQR or full range) for continuous variables will be presented.

A linear mixed model will be applied to compare the groups with respect to the primary outcome. The model will include baseline bedside (ie, clinic) diastolic BP (mean of the second and third bedside diastolic BPs, randomised group and minimisation factors (gestational age at the time of presentation with pre-eclampsia/gestational hypertension (continuous) and prescription of ACE inhibitor at randomisation) as fixed effects. Participant will be included as a random effect. For all participants included in the primary outcome analysis, the mean 24 hours average diastolic BP will be reported by randomised group. Adjusted mean differences between randomised groups with 95% CI and p value will be estimated from the model for the following comparison: self-management (intervention) versus usual care (control) at a single time point (V4: the final study visit).

Secondary BP outcomes will be analysed using the same method. Other secondary outcomes will be analysed using analysis of covariance to establish a covariant model to examine the effect of BP control in the postpartum period on cardiac structure and function, vascular function and cerebrovascular structure and function. If the model assumptions are not met and evidence of departure from normality is observed, transformations of the data will be employed or non-parametric tests will be carried out. Descriptive statistics (mean, SD, SE, range, etc) will also be calculated for each outcome for each group. Differences in the secondary outcomes will be compared between intervention and control groups.

Mean changes in BPs will be compared across the population and correlated with cardiovascular endpoints including cardiac structure and function reported from cardiac MRI and echocardiogram. Demographic and

**Box 1    Summary of the power calculations used to determine the trial sample size for the primary, and key secondary outcomes**

**Primary outcome measure**: 24-hour average diastolic blood pressure (mm Hg) at 6–9 months post partum as assessed by SPACELAB 90217 24 hours ambulatory blood pressure monitor

Sample size calculation: The detection of BP differences between the two arms of this trial is based on the mean diastolic blood pressure difference detected in the pilot SNAP-HT study at 6 months. The mean BP difference detected between the intervention and control arm at the 6-month time point was −4.5 mm Hg.[7] We have used a more conservative SD of 10 mm Hg in each arm (in SNAP-HT the SD was 8.2 mm Hg in the intervention arm and 9.8 mm Hg in the standard care arm) and a 10 mm Hg SD is in keeping with pooled SDs for ambulatory diastolic blood pressure readings from other studies. To detect a treatment effect on diastolic blood pressure of −4.5 mm Hg, powered to 80% at p=0.05 requires a total sample size of 158 and with 1:1 randomisation this would require 79 in each arm (total sample size of 158). During COVID-19 a Royal College of Obstetricians and Gynaecology guideline[46] was issued that recommended a home BP monitor be given to all women for the first week(s) after discharge. As a result of the potential dilution that self-monitoring in the control group could have, we recalculated our sample size. One systematic review[47] concluded self-monitoring in the control arm could lead to a potential 0.42 mm Hg dilution of the impact of self-management on diastolic BP, when measured using 24 hours ABPM. Therefore, assuming the same SDs of 10 mm Hg in each arm as in our original power calculation, we subtracted 0.42 mm Hg from the 4.5 mm Hg between group difference we had originally powered on. To remain powered at >80% and we would require 95 in each group (190 total) and hence we planned to over-recruit to 220 (rather than the original 200) to allow for this.

**Secondary outcome hypothesis: improved blood pressure control in the postpartum period in Physician Optimised Post-partum Hypertension Treatment (POP-HT) will result in improved cardiac, vascular and cerebrovascular phenotypes at 6–12 months post partum**

**Secondary outcome power calculations**

**(1) Cardiac structure**: Studies using echocardiography by our collaborators have compared BP and LV mass in pre-eclampsia patients and control patients, at 1-year post partum.[16 17] They found that a difference in BP at 1 year of 10 mm Hg in diastolic BP corresponded to significant differences in LV mass. SNAP-HT appeared to achieve a 50% reduction of anticipated BP difference seen between pre-eclamptic and normotensives at 1 year year by 6 months, that is, ~5 mm Hg. If it is assumed that the structural/phenotypic benefit results from the BP benefit, as we are hypothesising, then we must power to detect 50% of the phenotypic difference. In previous work by our group[45], we have demonstrated significant differences in LV mass/EDV (g/mL) in a similar age and predominantly female population with similar mean diastolic BP differences between groups to that seen in SNAP-HT. The LV mass/EDV (g/mL) in the group with high normal blood pressure was 1.54 g/ml g/mL vs 1.22 g/ml g/mL in those with optimal blood pressure with a standard deviationSD of 0.33 and 0.27, respectively, at p<0.001. Based on these assumptions, to observe a treatment effect of 0.16 (50% of the difference between 1.54 g/mL and 1.22 g/mL) on LV mass/EDV, requires 67 in the intervention arm and 67 in the control arm (132 total). This is calculated using the larger SD of 0.33 referenced above at a power of >80% to detect a difference between the groups at p=0.05. This number should take into account for the greater drop-out rate we may see for the MRI outcomes.

Continued

**Box 1    Continued**

**(2) Brain white matter integrity:** Work by our group[14], on pre-eclamptic pregnancy, showed an increased burden of temporal lobe white matter lesion volume 5–10 years after a pre-eclamptic pregnancy (23.2±13 µL) vs matched individuals who had a normotensive pregnancy (10.9±11.5 µL) at p<0.05. If we again assume we can detect a 50% of the phenotypic benefit with our intervention as outlined above, we would anticipate a 50% reduction in the burden of white matter lesions, that is, 6.15 µL (50% of 23.2–10.9 µL) in the intervention arm. With 71 in the intervention group and 71 in the control group (142 total), this will provide >80% power at p=0.05, even using the more conservative SD of 13 µL to detect a 50% improvement in white matter lesion volume between the intervention and the control group. This number should take into account for the greater dropout rate we may see for the MRI outcomes.

**(3) Aortic compliance**

Several studies assessing the impact of blood pressure on aortic compliance have shown that even modest reductions in systolic/diastolic blood pressure increase aortic distensibility/compliance. One such study had a mean difference in systolic blood pressure of 4.6 mm Hg between the two drug treatment arms at 52 weeks, akin to the same mean difference in SNAP-HT at 6 months, and other studies have suggested diastolic BP may be even more important in influencing aortic compliance. In this study with a mean 4.6 mm Hg difference in systolic BP there was a treatment difference of 0.12 (95% CI −0.35 to 0.60), p=0.60 in aortic compliance. Based on these assumptions, to observe a treatment effect from our intervention, with 100 in the intervention and 100 in the control arm we will be more than powered at >90% to detect a difference at p=0.05 in POP-HT. In this study, we will assess both aortic stiffness (by Vicorder PWV and AI) and aortic compliance by MRI although power calculation here is based on MRI measures of aortic compliance.

**Exercise ejection fraction**

Within our group, Huckstep *et al* compared resting and exercise ejection fractions for young adults with high normal BP versus a normotensive cohort.[48] The cohort was well-matched demographically to our planned study cohort, although it included both males and females. Resting ejection fraction (by Biplane Simpson's) was similar between groups but at 40%–60% of peak exercise intensity, the higher blood pressure group had a lower exercise ejection fraction than the normotensive cohort (73.9±3.25 vs 80.0%±4.54%, p<0.001) and in keeping with this, a smaller increase in ejection fraction when going from baseline to 40% exercise intensity (10.4±5.92 vs 19.0%±6.90%, p<0.001). Assuming the ~5 mm Hg mean BP improvement achieved in SNAP-HT again translates to a 50% phenotypic benefit, when assessing exercise ejection fraction we anticipate a 4.3% improvement in exercise stress ejection fraction in the intervention arm vs the control arm (4.3% is 50% of the difference, ie, 50% of 10.4%–19%). With 43 participants in the intervention arm and 43 in the control arm (86 total), we will be powered at >80% to detect such a difference at p=0.05. This calculation also takes account of the lower number likely to undertake the cardiopulmonary exercise testing at the final study visit.

physiological characteristics of the participants will be added to regression models as covariates to explore the determinants of change in BP comparing intervention and control groups.

## Analysis populations

The participants that will be included in the analysis will be all of those randomised. All data will be included

in the analysis as far as possible to allow full ITT analysis, though there will inevitably be the problem of missing data due to withdrawal, lost to follow-up or non-completion of questionnaire data.

### Decision points

There will be no formal interim analysis. The results once analysed will be reviewed by the research team, the TSC and Data and Safety Monitoring Committee (DSMC) and the PI/CI and other collaborators.

### The level of statistical significance

Level of significance will be tested as a 5% two-sided significant level.

### Procedure for accounting for missing, unused and spurious data

Missing data: Missing data will be reported with reasons given where available and the missing data pattern will be examined. We will explore the mechanism of missing data, though the mixed effects model implicitly accounts for data missing at random. The need for a sensitivity analysis taking into account missing data using multiple imputation will be considered and outlined further in the SAP. Spurious data will be assessed using standard editing criteria.

### Procedures for reporting any deviation(s) from the original SAP

The final statistical plan will be agreed prior to final data lock and prior to any analyses taking place. Any deviation thereafter will be reported in the final trial report.

### TRIAL OVERSIGHT

The trial will be conducted in accordance with the current approved protocol,GCP, relevant regulations and standard operating procedures. A risk assessment and monitoring plan are not being prepared before the study opens, as it is a low-risk intervention.

### Trial committees

A TSC will convene prior to the study starting and half-yearly thereafter to review and address key aspects of the study including the following:
1. Recruitment.
2. Safety/adverse events (SAEs) as defined in table 2 below.
3. Withdrawals.
4. Data management.
5. SAP.

The TSC will also function as a DSMC for this particular study and there will be a smaller trial management committee, which will focus more on the week-to-week running of the trial and will be on a more regular basis.

### Monitoring

Direct access will be granted to authorised representatives from the Sponsor within the appropriate department and host institution for monitoring and/or audit of the study to ensure compliance with regulations. Following written standard operating procedures, the monitoring visits will verify that the clinical trial is conducted and data are generated, documented and reported in compliance with the protocol, good clinical practice (GCP) and the applicable regulatory requirements.

### SAFETY REPORTING

### Procedures for reporting adverse events

A serious adverse event (SAE) occurring to a participant will be reported to the REC that gave a favourable opinion of the study where in the opinion of the Chief Investigator the event was 'related' (resulted from administration of any of the research procedures) and 'unexpected' in relation to those procedures. Reports of related and unexpected SAEs should be submitted within 15 working days of the Chief Investigator becoming aware of the event, using the HRA report of SAE form (see HRA website).

| Table 2 | Adverse event definitions |
| --- | --- |
| Serious adverse event (SAE) | An SAE is any untoward medical occurrence that:<br>► Results in death<br>► Is life-threatening<br>► Requires inpatient hospitalisation or prolongation of existing hospitalisation<br>► Results in persistent or significant disability/incapacity<br>► Consists of a congenital anomaly or birth defect.<br>Other 'important medical events' may also be considered an SAE when, based on appropriate medical judgement, the event may jeopardise the participant and may require medical or surgical intervention to prevent one of the outcomes listed above.<br>Note the term 'life-threatening' in the definition of 'serious' refers to an event in which the participant was at risk of death at the time of the event; it does not refer to an event, which hypothetically might have caused death if it were more severe. |

NB: To avoid confusion or misunderstanding of the difference between the terms 'serious' and 'severe', the following note of clarification is provided: 'Severe' is often used to describe intensity of a specific event, which may be of relatively minor medical significance. 'Seriousness' is the regulatory definition supplied above.

The severity of events will be assessed on the following scale: 1=mild, 2=moderate, 3=severe.

Non-serious AEs considered related to the trial intervention as judged by a medically qualified investigator or the sponsor will be followed up once the event is considered stable. It will be left to the investigator's clinical judgement to decide whether or not an AE is of sufficient severity to require the participant's removal from the trial. A participant may also voluntarily withdraw from the trial due to what he or she perceives as an intolerable AE. If either of these occurs, the participant must undergo an end of trial assessment and be given appropriate care under medical supervision until symptoms cease, or the condition becomes stable.

### Events exempt from immediate reporting as SAEs

There are a number of expected admissions/consultations with healthcare providers that will be expected take place as part of the natural history of pre-eclampsia and gestational hypertension during the trial period. These will be classed as 'Foreseeable Events' exempt from reporting as SAEs and include:

► Maternal morbidity: visual disturbance; pulmonary oedema; respiratory failure; myocardial ischaemia; hepatic dysfunction, hepatic haematoma or rupture; and acute kidney injury, or severe hypertension (>180mmHg systolic and/or >110mmHg diastolic).
► Postpartum haemorrhage.
► Lower genital tract bleeding.
► Sepsis.
► Admission to hospital for pre-eclampsia, monitoring of hypertension, or symptoms of low BP.
► Preplanned hospitalisation.
► Diagnostic and therapeutic procedures including blood transfusion.
► Worsening pruritus.
► A pre-existing maternal condition (such as renal disease), unless it causes increased clinical concern.
► Admission for psychiatric or social reasons.
► Retained placenta.
► Extended hospital stay of the mother due to the need to keep the baby/babies in hospital.
► Neonatal care unit admission for indications unrelated to pregnancy hypertension, such as neonatal hyperbilirubinaemia or unanticipated care for a fetal anomaly.
► Fetal congenital anomaly.

This list is not exhaustive and therefore any other 'minor medical significance symptom', as judged by the CI/PI, which does not require inpatient hospitalisation/prolongation of existing hospitalisation or result in persistent or significant disability/incapacity, and is not life-threatening and, does not result in death, will not be classed as an adverse event not an SAE.

### DISCUSSION

Until recently, key evidence missing from trials of self-monitoring/telemonitoring was whether it actually led to lower BP. In 2018, the TASMIN-H4 randomised trial[38]

showed that GPs using self-monitored BP to titrate anti-hypertensives, with or without telemonitoring, achieved better BP control for patients using telemonitoring. As with previous trials, the mechanism of action appeared to be medication optimisation. More recent work shows that patient and clinician experience was largely positive and cost-effectiveness analysis suggests that self-monitoring in this context is cost-effective by NICE criteria.[39] Self-monitoring can be combined with self-titration of medication, a process known as self-management. The SNAP-HT trial[1] demonstrated that self-management postpartum following a hypertensive pregnancy offers great promise. The purpose of the POP-HT trial is to assess whether this BP reduction can be reproduced in a larger, PROBE study (Prospective Randomized Open, Blinded End-point) powered to detect differences in BP as the primary outcome.

Our group have studied women 10 years after hypertensive and normotensive pregnancies to characterise their cardiovascular, vascular and cerebrovascular systems. Consistent with previous reports,[18 19 40] women who had been through hypertensive pregnancy were more likely to have left ventricular hypertrophy (LVH), increased LV mass and impaired diastolic function. In addition, cerebrovascular changes were evident including; lower grey matter volumes, and greater white matter lesion density compared with the control population.[14 41] Vascular phenotypic changes included reduced capillary density and increased aortic stiffness.[15 42] These phenotypic differences were not explained by differences in traditional cardiovascular risk factors at the time of assessment. It is possible such differences emerge during pregnancy and persist independent of postpartum BP variability but this is not really known[43] and, although the cardiovascular adaptations that emerge during hypertensive pregnancies reverse to some extent postpartum,[17–20] an alternative hypothesis is that long-term 'risk' reflects a failure of the cardiac, vascular or cerebral systems to 'normalise' after pregnancy.[21 22 43 44] The POP-HT trial aims to determine if these phenotypic changes emerge as early as 6–9 months and if so, whether they can be mitigated by postpartum BP optimisation. If so, this may offer one approach to modify long-term end organ changes,[45] in addition to any beneficial effects on BP that are demonstrated in this trial.

### ETHICS AND DISSEMINATION
#### Declaration of Helsinki

The investigator will ensure that this trial is conducted in accordance with the principles of the Declaration of Helsinki.

### PUBLICATION POLICY

The investigators will be involved in reviewing drafts of the manuscripts, abstracts, press releases and any other publications arising from the study. Authors will acknowledge that the study was funded by the British

Heart Foundation Clinical Research Training Fellowship (BHF Grant number FS/19/7/34148). Authorship will be determined in accordance with the ICMJE guidelines and other contributors will be acknowledged. The summarised results will be published in a scientific journal/s and summarised on the Cardiovascular Clinical Research Facility (CCRF) website for participants to read. Should participants wish to have a copy of any papers published, they merely need to contact the study team, using the contact details provided on their PIS, and the team would be happy to provide one.

## Guidelines for Good Clinical Practice

The investigator will ensure that this trial is conducted in accordance with relevant regulations and with Good Clinical Practice. The trial protocol and all accompanying documentation has been approved by the sponsor, an external REC, the HRA (Ethics Ref: 19/LO/1901; IRAS Project ID: 273353) and OUH (the local NHS trust) TMA.

## Consent

The participant must personally sign and date the latest approved version of the informed consent form(s) before any trial specific procedures are performed. Please see online supplemental appendix E for details of how consent will be modified during the COVID-19 pandemic to reduce risk of the paper acting as a vector for transmission of coronavirus. Full details on the consent process are in the fully study protocol.

## Participant confidentiality

The study will comply with the General Data Protection Regulation (GDPR) and Data Protection Act 2018, which require data to be de-identified as soon as it is practical to do so. Further details are outlined in the study protocol and participant information sheets.

## Access to data

Direct access will be granted to authorised representatives from the Sponsor, host institution and the regulatory authorities to permit trial-related monitoring, audits and inspections. Further detail is provided in the fully study protocol. Data will be made available, under certain circumstances, following data lock on request to the study PI.

## Data management

The study will comply with the GDPR and Data Protection Act 2018. The University of Oxford, as sponsor will act as data controller for the study.

## Author affiliations

[1]Division of Cardiovascular Medicine, University of Oxford, Oxford, UK
[2]Nuffield Department of Primary Care Health Sciences, University of Oxford, Oxford, UK
[3]Oxford Cardiovascular Clinical Research Facility, University of Oxford, Oxford, UK
[4]Radcliffe Department of Medicine, University of Oxford, Oxford, UK
[5]Institute of Biomedical Engineering, Department of Engineering Science, University of Oxford, Oxford, UK
[6]Department of Sport, Exercise and Health, University of Basel, Basel, Switzerland
[7]Nuffield Department of Women's and Reproductive Health, University of Oxford, Oxford, UK
[8]Obstetrics and Gynecology, University of London Saint George's, London, UK
[9]Women's Health Academic Centre, King's College London, London, UK
[10]Division of Cardiovascular Medicine, Radcliffe Department of Medicine, University of Oxford, Oxford, UK
[11]Department of Primary Care Health Sciences, University of Oxford, Oxford, UK

**Correction notice** The article has been corrected since it was published online. Co-author Maryam Khan's name has been updated to Miriam Lacharie.

**Collaborators** Department of Sport, Exercise and Health, University of Basel, Switzerland: Professor Henner Hanssen MD, Maternal-fetal Medicine Unit, St George's University Hospitals NHS Foundation Trust: Professor Basky Thilaganathan FRCOG, King's College London: Professor Lucy Chappell FRCOG.

**Contributors** JK, AJL, RJM, KLT, YK, CA, AF, AC, JM, LM and PL contributed to the design of the study. JK, AJL, RM and PL secured funding. JK, AJL, RJM, YK, AM, AF, WW, KS, LM, CA, AC, MS, CR, KLT, WL and PL refined the overall study protocol and lead the project delivery. BT and LC have provided guidance and external refinement. JK, MS and CR designed and oversee the delivery of the intervention. JK, AC, YK and WW will contribute to 24-hour BP data acquisition and analysis. AJL, WL, JK and RJM contributed to the development of the Brain and Cardiac MRI protocols. JK, RJM, ML, WL and AJL will contribute to MRI image acquisition and quality control. WL will lead brain MRI image processing and analysis. JK will lead the cardiac MRI analysis with support from AJL, ML and WW. EMT and BR helped develop the renal MRI sequences and will lead on renal MRI analysis. Echocardiography acquisition and analysis will be overseen by JK and performed by blinded study investigators. Cardiopulmonary exercise testing and peripheral cardiovascular risk assessment will be overseen by JK, AJL, WW, AC and PL. HH will oversee retinal image acquisition and analysis. AF will run the substudy on endothelial cells and, CT will oversee analysis of circulating biomarkers within the substudy alongside AF.

**Funding** The research is being financed by a British Heart Foundation Clinical Research Training Fellowship (BHF Grant number FS/19/7/34148).

**Competing interests** LM is supported by the NIHR Oxford Biomedical Research Centre and is a part-time employee of Sensyne Health PLC and holds shares in this company. RJM has received BP monitors for research from Omron and is working with them to develop a telemonitoring system. Any fees/consultancy from this work are paid to his institution.

**Patient consent for publication** Not applicable.

**Provenance and peer review** Not commissioned; externally peer reviewed.

**ORCID iDs**
Jamie Kitt http://orcid.org/0000-0003-3247-5126
Katherine Louise Tucker http://orcid.org/0000-0001-6544-8066
Basky Thilaganathan http://orcid.org/0000-0002-5531-4301
Richard J McManus http://orcid.org/0000-0003-3638-028X

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
