## [Reviewer comments · BMJ Open]

ARTICLE DETAILS

TITLE (PROVISIONAL)	Post-partum blood pressure self-management following hypertensive pregnancy: protocol of the Physician Optimised Post-partum Hypertension Treatment (POP-HT) trial
AUTHORS	Kitt, Jamie; Frost, Annabelle; Mollison, Jill; Tucker, Katherine; Suriano, Katie; Kenworthy, Yvonne; McCourt, Annabelle; Woodward, William; Tan, Cheryl; Lapidaire, Winok; Mills, Rebecca; Khan, Maryam; Tunnicliffe, Elizabeth M; Raman, Betty; Santos, Mauro; Roman, Cristian; Hanssen, Henner; Mackillop, Lucy; Cairns, Alexandra; Thilaganathan, Basky; Chappell, Dr Lucy; Aye, Christina; Lewandowski, Adam; McManus, Richard; Leeson, Paul

VERSION 1 – REVIEW

REVIEWER	Sharman, James Menzies Research Institute Tasmania
REVIEW RETURNED	24-May-2021

GENERAL COMMENTS	This is a comprehensive paper that presents the protocol for an RCT in the setting of post-partum blood pressure control for women with hypertensive disorders of pregnancy. Comments and suggestions are generally minor, although I think it is important to clarify some aspects related to the primary endpoint of 24 hour ambulatory DBP as mentioned below. The 2008 Lancet Viewpoint paper from Bryan Williams and colleagues draws out compelling evidence as to why SBP is the most important BP treatment target and less emphasis for various reasons should be placed on DBP. The viewpoint is based mostly from non-pregnant population data among older people but some of the arguments are potentially relevant. For example that SBP is harder to control with medication than DBP, thus targeting DBP will still leave many patients with uncontrolled SBP, but if targeting SBP then DBP is almost always controlled. Is this true in the setting of hypertension disorders of pregnancy, and should the authors provide more justification for the DBP target in this trial? The follow up period is invariably described as ‘six months’ or ‘six to nine months’ or ‘6-12 months’ post-partum. This could be amended for consistency throughout (text and figures/tables in the appendices). Appendix A, trial flow chart, potential confusion: why is 24hr BP monitoring listed as ‘optional’ at weeks 1 and 6 (visits 2 and 3)? Or is it that “(this is the only variable measured at V3).” Why is the word ‘optional’ used next to the primary endpoint?
--

	Appendix B, given that there are several types of BP measurement, it would be good to clarify at each mention. Currently, "BP measurement" is a procedure at all visits in both study arms, but "Fitting 24 hr BP monitor" is only at visit 3 and 4 – is this correct? If so, it does not coalesce with information presented elsewhere. On page 17 text is different again, where it is mentioned that 24 hr ABP measurement will be recorded at visit 3 and 'visit 4 (6 months (up to 12 months) post partum.' Please provide consistent clarity as to exactly when the primary endpoint is being measured. If only at one time point, please include the rationale for this. Connected to the questions above; for the primary outcome analysis (page 19), is this for the between-group difference for the change in 24 hour ambulatory DBP (change between two time points)? Unclear if it is a single time point analysis. The description of statistical methods (page 20) suggests there is a baseline value. Page 20, secondary power calculations for aortic compliance are not clear whether this is referring to vicorder aortic PWV (this is aortic 'stiffness' but is often used interchangeably with 'compliance' even though this is the inverse of stiffness) or MRI aortic compliance. Both are mentioned in different places. Could this secondary endpoint be clarified please.
--	--

REVIEWER	Denolle, Thierry Hôpital Arthur-Gardiner
REVIEW RETURNED	07-Aug-2021

GENERAL COMMENTS	Some feedbacks:  1) The dates of the study should be included in the manuscript: start and end of the study. 2) I do not find any comment on the modifications of the treatment with BP results with HBP device. How do you define normal BP: < 135/ 85 mmHg for HBP or another level? How the GP/ mid-wife will adjust the treatment? Is it a standardised modification for the treatment? It could be a bias for BP results. 3) Why is the number of drugs not a secondary objective at the different visits? It is very important to interpret the primary objective. 4) Why is the feasibility of this HBP with telemetry not also a secondary objective? 2 important limitations for this study:  1) the feasibility: only 20% loss to follow up is planned in this study, with several visits which are sometimes very long (up to 4 hours) with echo, MRI...! Are you sure that it is safe enough? Besides, how many hypertensive pregnant women are necessary to obtain 200 included patients? Is your recruitment process efficient enough to really achieve such goal? 2) I do not really understand the impact of the Covid 19 on the study protocol: it seems that the control group has also an HBP device. Don't you think that these women will measure also BP between the visits and that it will be a bias to interpret the results between the 2 arms?
--

VERSION 1 – AUTHOR RESPONSE

Reviewer #1:

Dr. James Sharman, Menzies Research Institute Tasmania

Comments to the Author:

This is a comprehensive paper that presents the protocol for an RCT in the setting of post-partum blood pressure control for women with hypertensive disorders of pregnancy. Comments and suggestions are generally minor, although I think it is important to clarify some aspects related to the primary endpoint of 24 hour ambulatory DBP as mentioned below.

The 2008 Lancet Viewpoint paper from Bryan Williams and colleagues draws out compelling evidence as to why SBP is the most important BP treatment target and less emphasis for various reasons should be placed on DBP. The viewpoint is based mostly from non-pregnant population data among older people but some of the arguments are potentially relevant. For example that SBP is harder to control with medication than DBP, thus targeting DBP will still leave many patients with uncontrolled SBP, but if targeting SBP then DBP is almost always controlled. Is this true in the setting of hypertension disorders of pregnancy, and should the authors provide more justification for the DBP target in this trial?

Response: We thank the reviewer for highlighting the commentary by Williams et al [1], which as the reviewer notes, relates to individuals aged 50 years or older who are almost exclusively non-pregnant. In those below the age of 50 years diastolic blood pressure has been shown to be a better correlate to long-term cardiovascular risk and is the predominant hypertensive pathophysiology in young adults [2]. Given all our patients are young females, below the age of 50 years, our research has focused on the impact of better blood pressure control on diastolic blood pressure. Our choice of diastolic blood pressure as the primary outcome was also supported by our pilot trial (SNAP-HT [3]) that demonstrated a self-management intervention resulted in significant diastolic blood pressure differences up to 4 years post-partum [4]. To better explain our choice of diastolic blood pressure we have added reference to the above points in the Introduction on pages 5-6.

The follow up period is invariably described as 'six months' or 'six to nine months' or '6-12 months' post-partum. This could be amended for consistency throughout (text and figures/tables in the appendices).

Response: We thank the reviewer for this and have clarified the manuscript.

Appendix A, trial flow chart, potential confusion: why is 24hr BP monitoring listed as 'optional' at weeks 1 and 6 (visits 2 and 3)? Or is it that "(this is the only variable measured at V3)." Why is the word 'optional' used next to the primary endpoint?

Response: We thank the reviewer for this comment and have duly reworded Appendix A to avoid confusion. The 24hr BP monitor is not part of the week 1 follow up (V2) but is a measure at week 6 (V3) and during the final visit (V4). The changes are tracked in bold in the amended supplementary file document containing Appendix A.

Appendix B, given that there are several types of BP measurement, it would be good to clarify at each mention. Currently, "BP measurement" is a procedure at all visits in both study arms, but "Fitting 24 hr BP monitor" is only at visit 3 and 4 – is this correct?

Response: We thank the reviewer for this comment, and have duly edited the wording in appendix B to clarify what type of BP measurement is being done at each study visit. The changes are tracked and highlighted in the amended supplementary file document containing Appendix B.

If so, it does not coalesce with information presented elsewhere. On page 17 text is different again, where it is mentioned that 24 hr ABP measurement will be recorded at visit 3 and 'visit 4 (6 months (up to 12 months) postpartum.' Please provide consistent clarity as to exactly when the primary endpoint is being measured. If only at one time point, please include the rationale for this.

Response: We thank the reviewer for flagging this confusion in the wording. 24hr ABPM is fitted at two time points in the trial – the visit 3 at week 6 and the final visit 4. The primary outcome is the 24hr diastolic blood pressure at visit 4 but 24hr blood pressure data is being collected earlier in the trial to allow additional longitudinal analysis of blood pressure changes post-partum. We have duly edited the wording in appendix B and on page 17 of the supplementary file to clarify this, as well as within the main protocol itself, where a paragraph has been added before the table of 'study aims and objectives' on page 8.

Connected to the questions above; for the primary outcome analysis (page 19), is this for the between-group difference for the change in 24 hour ambulatory DBP (change between two time points)? Unclear if it is a single time point analysis. The description of statistical methods (page 20) suggests there is a baseline value.

Response: We thank the reviewer for this comment and have updated the protocol wording on page 18 to align with the wording in the finalised SAP: the primary comparison is adjusted mean difference in mean 24 hour average diastolic blood pressure between randomised groups adjusted for baseline bedside (i.e. clinic) diastolic BP and minimisation factors (gestational age (continuous) and prescription of ACE inhibitor at randomisation) as fixed effects with participant as a random effect.

Page 20, secondary power calculations for aortic compliance are not clear whether this is referring to vicorder aortic PWV (this is aortic 'stiffness' but is often used interchangeably with 'compliance' even though this is the inverse of stiffness) or MRI aortic compliance. Both are mentioned in different places. Could this secondary endpoint be clarified please.

Response: We thank the reviewer for this very insightful comment. We have duly edited the study aims and objectives table (page 9 of the protocol paper) and page 17 within the power calculations table to make it clear that we are assessing aortic stiffness (PWV) at baseline and 6-12 months, as well as aortic compliance (by MRI) at a single timepoint during the V4 imaging visit at 6-12 months

Reviewer #2:

Dr. Thierry Denolle, Hôpital Arthur-Gardiner

Comments to the Author:

1) The dates of the study should be included in the manuscript: start and end of the study

Response: We thank the reviewer for this comment and have included the following dates on page 6:

Planned Trial Period 31/12/19-01/12/30

Planned Recruitment period 31/12/19- 31/08/21

2) I do not find any comment on the modifications of the treatment with BP results with HBP device. How do you define normal BP: < 135/ 85 mmHg for HBP or another level? How the GP/ mid-wife will adjust the treatment? Is it a standardised modification for the treatment? It could be a bias for BP results.

Response: Thank you for this comment. The information on how blood pressure is modified was contained in pages 7-10 of appendix D in the supplementary file. We now more effectively signpost this supplement in the manuscript (on page 7) and have also made clearer in the main text that titration is triggered when BP is consistently <130 mmHg systolic and <80 mmHg diastolic. In addition, we have made clearer in this section (on page 7) that GP/midwife led blood pressure adjustment will also be determined by the UK guidance (NICE NG133) on blood pressure modification after pregnancy.

3) Why is the number of drugs not a secondary objective at the different visits? It is very important to interpret the primary objective.

Response: We thank the reviewer for this comment. This data is collected as part of the study and will be reported in the trial. We have added reference to this in the main paper to page 13 'assessments during study visits'.

4) Why is the feasibility of this HBP with telemetry not also a secondary objective?

Response: The feasibility of HBP with telemetry in this groups was reported in our previous pilot RCT (SNAP-HT) [3] whereas POP-HT is powered to test for differences in ambulatory diastolic blood pressure between study arms. Nevertheless, we are also collecting relevant data to assess feasibility in the current trial and this has been made clearer in the 'study aims and objectives table' on page 10. Intervention compliance data is also automatically recorded as part of the telemetry system for all those in the intervention arm.

2 important limitations for this study:

1) the feasibility: only 20% loss to follow up is planned in this study, with several visits which are sometimes very long (up to 4 hours) with echo, MRI...! Are you sure that it is safe enough? Besides, how many hypertensive pregnant women are necessary to obtain 200 included patients? Is your recruitment process efficient enough to really achieve such goal?

Response: Whilst we appreciate the concern of the reviewer, whilst this manuscript has been under review, the study has been recruiting and we have already reached the target of >200 participants. Of these, 180 participants have now completed visit 4, of which 13 withdrew and 2 were lost to follow up (15/180 – 8%). The remainder of these 180 have undergone the full study protocol. We remain confident we will achieve our planned recruitment and loss to follow up targets for those remaining in the study.

2) I do not really understand the impact of the Covid 19 on the study protocol: it seems that the control group has also an HBP device. Don't you think that these women will measure also BP between the visits and that it will be a bias to interpret the results between the 2 arms?

Response: The reviewer has highlighted an important point. The updated Royal College of Obstetrics & Gynaecology (RCOG) guidance published in response to the COVID-19 pandemic [5] advocated home blood pressure monitoring on discharge for all women who had had a hypertensive pregnancy to minimise hospital attendance and home visits. This advice was adopted by our local hospital in July 2020 and we were required to include this as standard care to continue the study during the pandemic. In practice, the use of home monitors was variable, and standard care did not include the dose titration we provided in the intervention arm. Nevertheless, to account for a potential dilution effect from provision of home monitors we re-powered the sample size based on systematic review [6] evidence of the likely impact of self-monitoring alone on differences in 24hr diastolic blood pressure (0.42mmHg). To remain powered at > 80% we required 95 in each group (190 total) and hence we planned to over-recruit to 220 (rather than the original 200). We have realised the power calculation

included in the manuscript is the original power calculation and does not reflect the power calculation contained in our finalised statistical analysis plan. We have therefore updated the relevant section of the paper (pages 16-17).

References

1. Williams B, Lindholm LH, Sever P. Systolic pressure is all that matters. *Lancet*. 2008 Jun 28;371(9631):2219-21. doi: 10.1016/S0140-6736(08)60804-1. Epub 2008 Jun 16. PMID: 18561995.
2. Luo D, Cheng Y, Zhang H, Ba M, Chen P, Li H et al. Association between high blood pressure and long term cardiovascular events in young adults: systematic review and meta-analysis *BMJ* 2020; 370 :m3222 doi:10.1136/bmj.m3222
3. Cairns AE, Tucker KL, Leeson P, Mackillop LH, Santos M, Velardo C, Salvi D, Mort S, Mollison J, Tarassenko L, McManus RJ; SNAP-HT Investigators. Self-Management of Postnatal Hypertension: The SNAP-HT Trial. *Hypertension*. 2018;72(2):425-432
4. Short-Term Postpartum Blood Pressure Self-Management and Long-Term Blood Pressure Control: A Randomized Controlled Trial. Jamie A. Kitt, Rachael L. Fox, Alexandra E. Cairns, Jill Mollison, Holger H. Burchert, Yvonne Kenworthy, Annabelle McCourt, Katie Suriano, Adam J. Lewandowski, Lucy Mackillop, Katherine L. Tucker, Richard J. McManus, Paul Leeson *Hypertension*. 2021; 78:469–479
5. RCOG guidance: Guidance for maternal medicine in the evolving coronavirus (COVID-19) pandemic. Version 1. Published 30/03/2020 (updated throughout the pandemic and currently published version is 2.4: Published Friday 10 July 2020)
6. Sheppard, J.P., et al., Self-monitoring of Blood Pressure in Patients With Hypertension-Related Multi-morbidity: Systematic Review and Individual Patient Data Meta-analysis. *Am J Hypertens*, 2020. 33(3): p. 243-251.

VERSION 2 – REVIEW

REVIEWER	Denolle, Thierry Hôpital Arthur-Gardiner
REVIEW RETURNED	26-Oct-2021
GENERAL COMMENTS	I thank the authors for their answers and modifications.